# Decoupled Kernel Neural Processes: Neural-Network-Parameterized Stochastic Processes Using an Explicit Data-driven Kernel

## Abstract

Neural Processes (NPs) are a class of stochastic processes parametrized by neural networks. Unlike traditional stochastic processes (*e.g.*, Gaussian processes), which require specifying explicit kernel functions, NPs implicitly learn kernel functions appropriate for a given task through observed data. While this data-driven learning of stochastic processes has been shown to model various types of data, the current NPs' implicit treatment of the mean and the covariance of the output variables limits its full potential when the underlying distribution of the given data is highly complex. To address this issue, we introduce a new class of neural stochastic processes, Decoupled Kernel Neural Processes (DKNPs), which explicitly learn a separate mean and kernel function to directly model the covariance between output variables in a data-driven manner. By estimating kernel functions with cross-attentive neural networks, DKNPs demonstrate improved uncertainty estimation in terms of conditional likelihood and diversity in generated samples in 1-D and 2-D regression tasks, compared to other concurrent NP variants. Also, maintaining explicit kernel functions, a key component of Gaussian processes, allows the model to reveal a deeper understanding of underlying distributions.

## 1 Introduction

Neural processes (NPs) (Garnelo et al., 2018a;b) are a class of stochastic processes parametrized by neural networks. By embracing statistical properties in stochastic processes, NPs can effectively estimate the uncertainty of underlying distributions of functions with a set of realizations and their data points. Different from traditional stochastic processes (*e.g.*, Gaussian processes (GP) (Rasmussen & Williams, 2006)), NPs learn data-driven stochastic processes without a need to specify or keep an explicit form of kernel functions. As a result of their simplicity and flexibility, there have been numerous efforts to further develop improved variants of NPs (Kim et al., 2019; Lee et al., 2020; Gordon et al., 2020) and apply them to various downstream tasks (Singh et al., 2019; Requeima et al., 2019).

Though significant progress has been made in NPs, the current architectures of NPs either fails to capture output dependencies as in Conditional NPs (Garnelo et al., 2018a; Gordon et al., 2020), or indirectly capture the full stochasticity present in the traditional stochastic processes. For instance, different from GPs, conventional NPs reserve stochasticity in a global latent variable and output variables separately. The output variables estimate point-wise uncertainty, which corresponds to the diagonal elements of a kernel matrix. Similarly, the global latent variable takes charge of the functional uncertainty and diversity, represented by the full covariance matrix in GPs. Due to this inductive bias of conventional NPs, the role of estimating functional stochasticity is mainly assigned to a fixed-length vector (*i.e.* the global latent variable), and consequently, it becomes challenging to capture the underlying distributions in complex scenarios (*e.g.*, variable relationships are periodic or abruptly changing at a certain point). Although several approaches attempt to alleviate the problem by introducing attention (Kim et al., 2019) and bootstrapping (Lee et al., 2020) on top of conventional NPs, the problem still exists as the architectural limitation (*i.e.* implicit modeling of

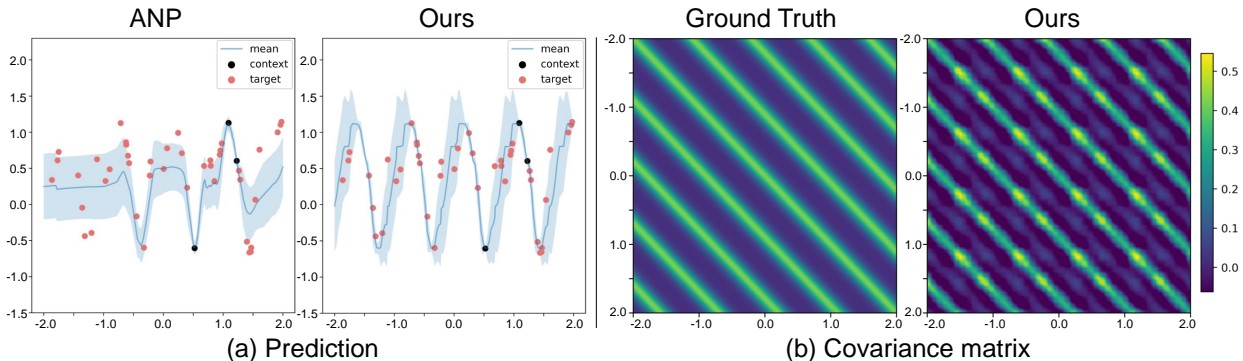

Figure 1: **(a)** Comparison of predictions with given context points (black dot) by the Attentive NP (ANP) and DKNP (Ours) after training them with samples generated from a periodic kernel with fixed hyperparameters. The mean (blue curve) and sigma (shaded blue area) predicted from the DKNP better represent the data, including target points (red dot) compared to the ANP. **(b)** Visualization of kernel functions learned by the DKNP. As a result of kernel learning in the DKNP, in this case, periodic prior distribution of data can be inferred.

the mean and covariance) has not been addressed directly. Besides this, as NPs implicitly learn the kernel functions inside the model, the interpretability of kernels such as in GPs (Lloyd et al., 2014) is diminished.

To address this concern, we propose *Decoupled Kernel Neural Processes* (DKNPs), a new class of neural stochastic processes that explicitly learn a separate mean and kernel function to directly model the covariance between output variables in a data-driven manner. Our experiments in 1-D and 2-D regression tasks reveal that the DKNP outperforms concurrent NP variants in terms of predictive likelihood. Especially, DKNPs effectively model the distributions having complex kernel functions (e.g., periodic function) contrary to existing NP variants as demonstrated in Figure 1(a). Also, as visualized in Figure. 1(b), DKNP achieves improved interpretability via explicitly learned kernels and better global coherence of generated samples.

## 2 Background

### 2.1 Neural Process

Given a function from a stochastic process observed in $n$ points, let us denote the input and output as $X = \{\boldsymbol{x}_i\}_{i=1}^n$ and $Y = \{\boldsymbol{y}_i\}_{i=1}^n$, respectively, where $\boldsymbol{x}_i \in \mathbb{R}^{d_x}$ and $\boldsymbol{y}_i \in \mathbb{R}^{d_y}$. For a set of target inputs $X_T = \{\boldsymbol{x}_i\}_{i \in T} \subset X$, NPs model the conditional distribution of target outputs $Y_T$ conditioned on the context set $(X_C, Y_C) = \{(\boldsymbol{x}_i, \boldsymbol{y}_i)\}_{i \in C}$ using a factorized Gaussian distribution:

$$\log p(Y_T | X_T, X_C, Y_C) = \sum_{i \in T} \log p(\boldsymbol{y}_i | \boldsymbol{x}_i, X_C, Y_C). \tag{1}$$

For obtaining the predictive distribution $p(\boldsymbol{y}_i | \boldsymbol{x}_i, X_C, Y_C)$, NPs use an encoder-decoder architecture that ensures the permutation invariance of the predictions of the target points given the context set $(X_C, Y_C)$.

Following Kim et al. (2019) and Lee et al. (2020), we consider the NP encoder consisting of two separate paths, namely the deterministic path and the latent path. For the deterministic path, $f_\theta$ represents each context point in $\{(\boldsymbol{x}_i, \boldsymbol{y}_i)\}_{i \in C}$ as $\boldsymbol{r}_i \in \mathbb{R}^{d_r}$, *i.e.*, $\boldsymbol{r}_i = f_\theta(\boldsymbol{x}_i, \boldsymbol{y}_i)$. Then, we aggregate the $\boldsymbol{r}_i$'s by averaging them across all context points, $\boldsymbol{r}_C = \frac{1}{n_c} \sum_{i \in C} \boldsymbol{r}_i$ where $n_c = |C|$. This vector $\boldsymbol{r}_C \in \mathbb{R}^{d_r}$ is the summarized representation of context points, and it is permutation invariant over the order of $(\boldsymbol{x}_i, \boldsymbol{y}_i) \in (X_C, Y_C)$.

The latent path of the NP encoder operates in a similar fashion to the deterministic path, *i.e.*, $\boldsymbol{e}_C = \frac{1}{n_c} \sum_{i \in C} \boldsymbol{e}_i$ where $\boldsymbol{e}_i = g_\phi(\boldsymbol{x}_i, \boldsymbol{y}_i)$. Unlike the deterministic path, however, the latent path uses stochastic layers for obtaining a distribution of the latent variable $\boldsymbol{z} \in \mathbb{R}^{d_z}$; $q(\boldsymbol{z}|\boldsymbol{e}_C) = \mathcal{N}(\boldsymbol{z}; \boldsymbol{\mu}_z, \boldsymbol{\sigma}_z^2)$ where $\boldsymbol{\mu}_z, \boldsymbol{\sigma}_z$ are the output of the additional fully-connected layer applied to $\boldsymbol{e}_C$. Finally, by concatenating these aggregated vectors $\boldsymbol{r}_C, \boldsymbol{z}$ with the target inputs $\boldsymbol{x}_i \in X_T$, the decoder $h_\psi$ produces the predictive distribution

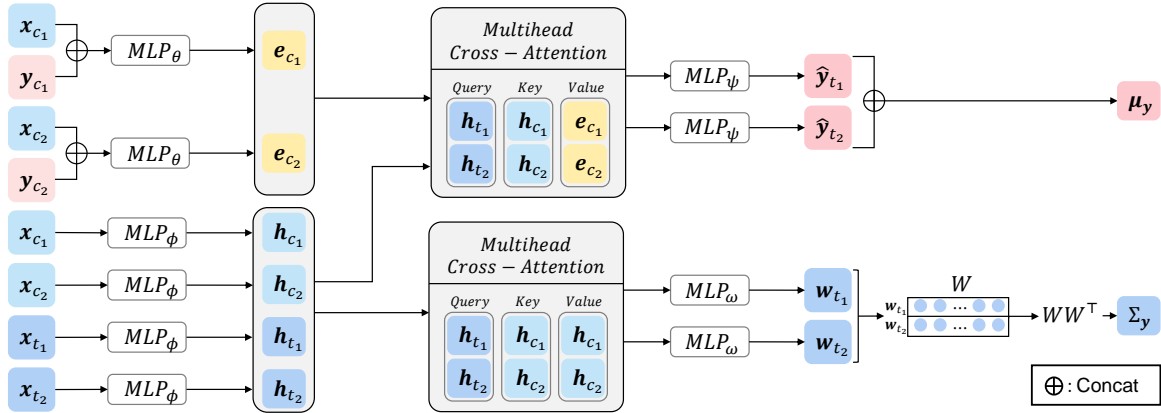

Figure 2: Model architecture of DKNPs. DKNPs estimate the predictive distribution $\mathcal{N}(y; \boldsymbol{\mu}_y, \Sigma_y)$ by using two attention-based deterministic paths using Multihead Cross-Attention (MCA) to model the mean vector (upper path) and the full covariance matrix (lower path). Note that the main difference between DKNPs and ANP (*i.e.,* another attention-based NP variant) is the lower path that estimates the covariance matrix by utilizing $x$ values of context points and target points.

$p(\boldsymbol{y}_t|\boldsymbol{x}_t, \boldsymbol{r}_C, \boldsymbol{z}) = \mathcal{N}(\boldsymbol{y}_t; \boldsymbol{\mu}_y, \boldsymbol{\sigma}_y^2)$ where $(\boldsymbol{\mu}_y, \boldsymbol{\sigma}_y) = h_\psi(\boldsymbol{x}_t, \boldsymbol{r}_C, \boldsymbol{z})$. Note that the $\boldsymbol{\sigma}_y$ is used for capturing the point-wise uncertainty.

During the training phase, the parameters are learned by maximizing the evidence lower bound of $\log p(Y_T|X_T, X_C, Y_C)$ via the reparametrization trick (Kingma & Welling, 2014; Rezende et al., 2014) as follows:

$$\log p(Y_T|X_T, X_C, Y_C) \geq \mathbb{E}_{q(\boldsymbol{z}|X_T, Y_T)}[\log p(Y_T|X_T, \boldsymbol{z})] - KL(q(\boldsymbol{z}|\boldsymbol{e}_T)\|q(\boldsymbol{z}|\boldsymbol{e}_C)) \tag{2}$$

where $q(\boldsymbol{z}|\boldsymbol{e}_T) = q(\boldsymbol{z}|X_T, Y_T)$ is a distribution of latent variable by encoding the set of target points $(X_T, Y_T) = \{(\boldsymbol{x}_i, \boldsymbol{y}_i)\}_{i \in T}$.

This objective function consists of two parts: 1) the reconstruction loss for the target points and 2) the KL divergence term, minimizing the divergence between two distributions $q(\boldsymbol{z}|X_T, Y_T)$ and $q(\boldsymbol{z}|X_C, Y_C)$. Note that, in practice, we assume that $X_C \subset X_T$ during the training phase. The KL divergence term encourages two distributions inferred by the context sets and target sets to be similar, which is reasonable because two sets are generated from the same function. Therefore, during the inference phase, the distribution $q(\boldsymbol{z}|X_C, Y_C)$ captures the functional stochasticity which is demonstrated with the coherent sample generation. It can be thought that $\boldsymbol{z}$ learns to capture the correlation of output variables of the stochastic processes.

## 2.2 Attentive Neural Processes

**Multihead Attention**    Given $n$ key-value pairs of matrices $K \in \mathbb{R}^{n \times d_{model}}$ and $V \in \mathbb{R}^{n \times d_{model}}$, and $m$ queries $Q \in \mathbb{R}^{m \times d_{model}}$, where $d_{model}$ is a hyperparameter, the scaled dot product attention is formulated as:

$$\text{Attention}(Q, K, V) = \text{Softmax}\left(\frac{QK^\top}{\sqrt{d_{model}}}\right)V \tag{3}$$

where $K$, $V$, and $Q$ are projected by learnable linear maps $W_s^K$, $W_s^V$, and $W_t^Q$ from the source $S$ and target $T$ and Softmax denotes a row-wise softmax function.

The attention mechanism can be calculated from multiple subspaces, namely, multihead attention (MHA) (Vaswani et al., 2017). Denoting a single attention head as $Head_i = Attention(Q_i, K_i, V_i)$, the aggregate attention from multiple subspaces can be expressed as:

$$\text{MHA}(Q, K, V) = \text{Concat}(Head_1, ..., Head_H)W^O \tag{4}$$

where $W^O$ is the learnable linear map for aggregating the subspaces.

| Methods | RBF | Periodic | Matérn 5/2 |
|---------|-----|----------|------------|
| CNP | 0.695 ($\pm$0.010) | -0.328 ($\pm$0.032) | 0.558 ($\pm$0.006) |
| NP | 0.577 ($\pm$0.015) | -0.619 ($\pm$0.005) | 0.417 ($\pm$0.009) |
| BNP | 0.754 ($\pm$0.004) | -0.018 ($\pm$0.042) | 0.617 ($\pm$0.006) |
| ANP | 1.086 ($\pm$0.001) | 0.831 ($\pm$0.011) | 1.020 ($\pm$0.000) |
| BANP | 1.084 ($\pm$0.001) | 0.821 ($\pm$0.018) | 1.018 ($\pm$0.001) |
| Ours | **1.109** ($\pm$0.001) | **0.941** ($\pm$0.006) | **1.039** ($\pm$0.004) |
| Oracle GP | 1.302 ($\pm$0.000) | 1.329 ($\pm$0.000) | 1.264 ($\pm$0.000) |

Table 1: Results (log-likelihood) of the DKNP, NP variants and GP Oracle in multiple 1D regression tasks. Note that to measure the results of DKNPs and NP variants, we use a single model for all test samples generated from GP with different kernel hyperparameters. In the case of Oracle GPs, we utilize a GP model corresponding to each test sample (*i.e.,* a GP model where the test data is sampled from) and report the averaged value.

**Attentive NPs**  Attentive NPs (ANPs) leverage attention to resolve the underfitting issue in NPs. Unlike NPs that produce a single variable $\boldsymbol{r}_C$ from the deterministic path, ANPs utilize a query-specific variable $\boldsymbol{r}_i^*$ by applying the attention score $a_i$ for each $\boldsymbol{r}_i$ during the aggregation of the deterministic path, formulated as $\boldsymbol{r}_i^* = \sum_{i \in C} a_i \cdot \boldsymbol{r}_i$. The attention-based aggregation of ANPs resembles how the GPs utilize the correlation to estimate the predictive distribution of the context and the target points.

## 3  Decoupled Kernel Neural Processes

Decoupled Kernel Neural Processes use attention to explicitly learn a separate mean and kernel function so as to directly model covariances between output variables with related input variables in a data-driven fashion, which is contrary to conventional NPs that implicitly model the mean and kernel function through the latent variable $\boldsymbol{z}$. As shown in Figure 2, DKNPs estimate the predictive distribution as multivariate Gaussian $\mathcal{N}(y; \boldsymbol{\mu}_y, \Sigma_y)$ by using two attention-based deterministic paths using Multihead Cross-Attention (MCA) to model the mean vector (upper path) and the full covariance matrix (lower path). Here, attention modules are extensively utilized as Le et al. (2018); Kim et al. (2019) have demonstrated attention was helpful in achieving low predictive uncertainty near the context points. With the predictive distribution $\mathcal{N}(y; \boldsymbol{\mu}_y, \Sigma_y)$, DKNPs are trained and evaluated based on the function likelihood.

The core design of DKNPs is motivated by the predictive posterior distribution of GPs for regression task, where $X_T, X_C, Y_C$ are used for deriving the posterior mean,[1] but only $X_T, X_C$ for deriving the posterior covariance as follows:

$$\textbf{GP regression:} \quad p(Y_T|X_T, X_C, Y_C) = \mathcal{N}\Big(Y_T; \Sigma_{X_C,X_T}^\top \Sigma_{X_C,X_C}^{-1} Y_C, \Sigma_{X_T,X_T} - \Sigma_{X_C,X_T}^\top \Sigma_{X_C,X_C}^{-1} \Sigma_{X_C,X_T}\Big), \quad (5)$$

$$\textbf{DKNP:} \quad p(Y_T|X_T, X_C, Y_C) = \mathcal{N}\Big(Y_T; \text{MLP}_\psi \circ \text{MCA}\big(\text{MLP}_\theta(X_C, Y_C), \text{MLP}_\phi(X_T, X_C)\big),$$

$$\text{MLP}_\omega \circ \text{MCA}\big(\text{MLP}_\phi(X_T, X_C)\big)\Big), \quad (6)$$

where, in contrast to GPs, DKNPs' covariance is learned via attention.[2] This decoupled process allows DKNPs to explicitly learn the prior of the given dataset, and thus act as a true generative process, ensuring the global consistency of all points in the given stochastic process samples. Unlike DKNPs, NPs pack all information $(X_C, Y_C, X_T)$ into latent variables to derive both mean and covariance, thus inherently becoming a conditional process that requires a sufficient amount of context points, unable to explicitly learn a prior.

Specifically, the DKNPs pass each context point $\{(\boldsymbol{x}_i, \boldsymbol{y}_i)\}_{i \in C}$, the concatenation of $\boldsymbol{x}_i$ and $\boldsymbol{y}_i$, to $\text{MLP}_\theta$ and represent it as $\boldsymbol{e}_i$. Similarly, we produce the representation vector of $\boldsymbol{x}_i$, $\boldsymbol{h}_i$, using another $\text{MLP}_\phi$ for all

---

[1]We follow a typical GP formulation where the mean function is set to zero.
[2]Note that DKNPs assume $C \subset T$ during training, which makes Eq. 5 and Eq. 6 technically different. This difference comes from DKNPs using all data points for better learning the kernels, unlike fixed-kernel GPs. DKNPs and GPs, however, are different methods and superior empirical performance led to the current design choice.

inputs $\{\boldsymbol{x}_i\}_{i \in T}$. Then, $\boldsymbol{e}_i$ and $\boldsymbol{h}_i$ are passed to the MCA module to create the mean vector $\boldsymbol{\mu}_y$. All heads in the MCA module perform cross-attention, $Q = \{\boldsymbol{h}_i\}_{i \in T}$, $K = \{\boldsymbol{h}_i\}_{i \in C}$, and $V = \{\boldsymbol{e}_i\}_{i \in C}$. We adopted the architecture of MCA used in image transformers (Parmar et al., 2018), where the original query vectors are added to the output from the MCA through the residual path. This allows the model to infer the output distributions without context points, which can be considered as the learned prior distributions of DKNPs. After the MCA, the last $\text{MLP}_\psi$ generates the predictive mean vector $\boldsymbol{\mu}_y$ for each data point. Intuitively, this can be interpreted as predicting the target mean based on the context and the correlation, which resembles the estimation of the predictive distribution in GPs and ANPs.

Unlike the NPs, DKNPs explicitly capture the correlation between the output variables using another multihead cross-attention (MCA), where $Q = \{\boldsymbol{h}_i\}_{i \in T}$, $K = \{\boldsymbol{h}_i\}_{i \in C}$, and $V = \{\boldsymbol{e}_i\}_{i \in C}$. Then $\text{MLP}_\omega$ produces the representation vector $\boldsymbol{w}_i \in \mathbb{R}_w^d$ for each position, which are combined to generate the covariance matrix $\Sigma = WW^\top$, where $W_{i,:} = \boldsymbol{w}_i$ and $\Sigma_{ij} = \text{kernel}(\boldsymbol{x}_i, \boldsymbol{x}_j) = \boldsymbol{w}_i^\top \boldsymbol{w}_j$. Note that since MCA leverages target points as query and context points as key and value, the representation vector $\boldsymbol{w}_i$ of the $i$-th target point is calculated based on the set of context points and the $i$-th target point only. Additionally, calculating $\Sigma_{ij}$ only requires the corresponding $\boldsymbol{w}_i$ and $\boldsymbol{w}_j$. That is, $\Sigma_{ij}$ remains the same regardless of other $k$-th target points where $k \neq i$ and $k \neq j$. Therefore, given the context points, DKNPs satisfy the consistency under marginalization.

One might consider using self-attention to let the model learn the correlation between all data points. However, the self-attention module on only $X$ as inputs receives no indication of context and target points and therefore fails to reduce the uncertainty near the points that have high confidence (*e.g.*, context points). Also, the interaction between the target points through self-attention does not guarantee consistency under the marginal distribution of target points when the context points are given. Lastly, it is also important to note that the representation $h$ is shared when modeling both the mean and the covariance. This motivation is drawn from Equation 5 that the calculation of the mean is also based on the kernel matrices, $\Sigma_{X_C, X_T}^\top$ and $\Sigma_{X_C, X_C}^{-1}$.

To train the DKNPs, the obtained mean vector $\boldsymbol{\mu}$ and the covariance matrix $\Sigma$ act as parameters of a predictive distribution $\mathcal{N}(Y; \boldsymbol{\mu}_Y, \Sigma_Y)$. Instead of maximizing the lower bound of the log-likelihood as in most NP models, the training objective of DKNPs is to maximize the tractable log-likelihood of the Gaussian as follows:

$$\log p(Y_T | X_T, X_C, Y_C) = \log \mathcal{N}(Y_T; \boldsymbol{\mu}_Y, \Sigma_Y) \quad \text{where} \quad \Sigma_Y = WW^\top. \tag{7}$$

Although the proposed objective function is equivalent to CNP's (Garnelo et al., 2018a), modeling the correlation between output variables for capturing functional stochasticity shares the same motivation of NPs, thus DKNP being one of NP variants. Further details and discussion about computational complexity of DKNPs can be found in Appendix B.

## 4 Experiments

We compare DKNP with diverse NP variants such as Conditional NP (CNP)(Garnelo et al., 2018a), NP, Bootstrapping NP (BNP)(Lee et al., 2020), ANP, and Bootstrapping ANP (BANP) on both 1D and 2D regression tasks. For a fair comparison, all NP variants use two paths for context encoding. CNP, BNP, and BANP have two deterministic paths, and NP and ANP have one deterministic and one stochastic path. Also, ANP and BANP have additional self-attention in both stochastic and deterministic paths. Following Lee et al. (2020), BNP and BANP were trained with 50 bootstrap context samples and tested with 4 samples. Note that we mostly followed the same hyperparameter setup used in Lee et al. (2020) and the details are described in Appendix A. Unlike the NP variants (except CNP) which are trained with the lower bound of conditional log-likelihood $\log p(Y_T | X_T, X_C, Y_C)$ or its slight modification, DKNP can directly evaluate it as well as maximize it during training. All results are also reported with the conditional log-likelihood to evaluate the methods' ability to model the given stochastic processes. Following (Le et al., 2018; Lee et al., 2020), we use importance weighting estimation (Burda et al., 2016) with 50 samples to evaluate the performance of NP variants that utilize $z$.

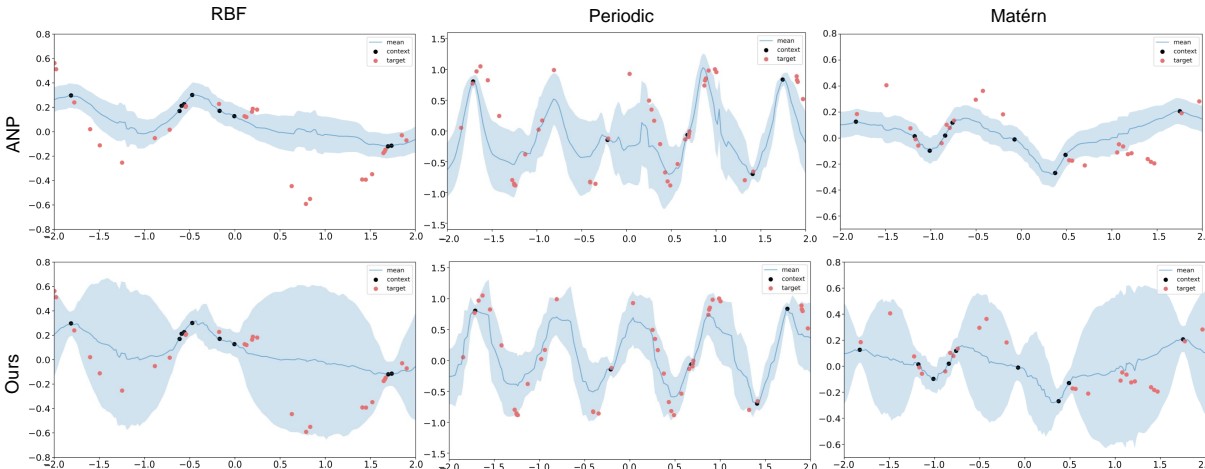

Figure 3: Comparison of predictions by ANP and DKNP. The dark blue line indicates the mean, and the light blue shade indicates one standard deviation range.

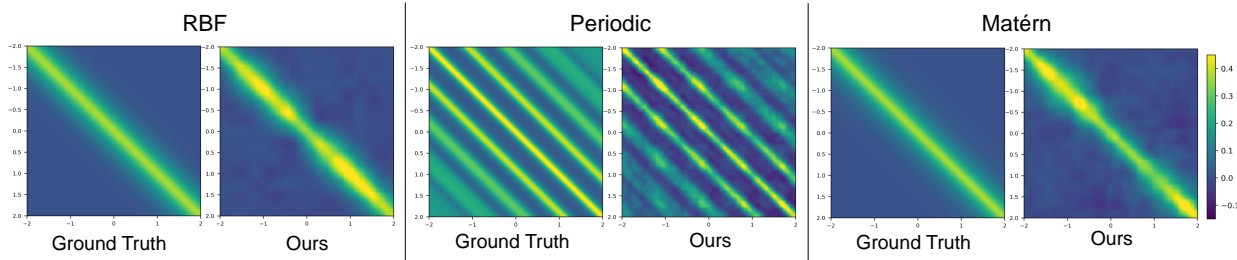

Figure 4: Visualization of the correlation matrices learned from the DKNP compared to the ground truths. We take a Monte Carlo estimate (averaging 1,000 kernel functions of the same type) to visualize the ground truth kernels for multiple hyperparameters at once.

### 4.1 1D Regression with Gaussian Processes Data

We first conduct basic evaluation of DKNP in comparison to NP variants by testing its ability to model 1D stochastic process samples generated from Gaussian Processes of diverse kernels, namely RBF, Periodic and Matérn $5/2$. In this task, we randomly sampled input $x$ from [-2, 2] and generated $y$ from GP kernels where the hyperparameters were also sampled both during training and testing. For example, in the case of RBF kernel, $k(x, x') = s^2 \exp(-||x - x'||^2/2l^2)$, we randomly sampled $s \sim \text{Unif}(0.1, 1.0)$ and $l \sim \text{Unif}(0.1, 0.6)$ at each iteration during the training and test phase. The number of context points $n_c$ is sampled from $Unif(3, 25)$ and the target points is $n_c + Unif(3, 50 - n_c)$. Details about the data generation process (kernel hyperparameters), the training process, and the evaluation process are in Appendix C.1.

Additionally, we report the results of Oracle GP to provide the upper-bound performance of the log-likelihood. Note that the results of DKNPs and NP variants are measured with a single model on all the test samples which are generated from several GP with different kernel hyperparameters. In the case of Oracle GP, however, a GP model where each test data is sampled from is utilized to measure the log-likelihood on the corresponding test data, and we report the averaged value.

The results in Table 1 show that DKNP can better model the given stochastic processes compared to various NP variants in terms of the log-likelihood. Note that the attention mechanism dramatically increases all methods' ability to model GP samples, as can be seen by the clear divide between CNP, NP, BNP and the rest. The true merit of DKNP, however, is its capability to better estimate the uncertainty of the stochastic processes thanks to its explicit modeling of the full covariance matrix.

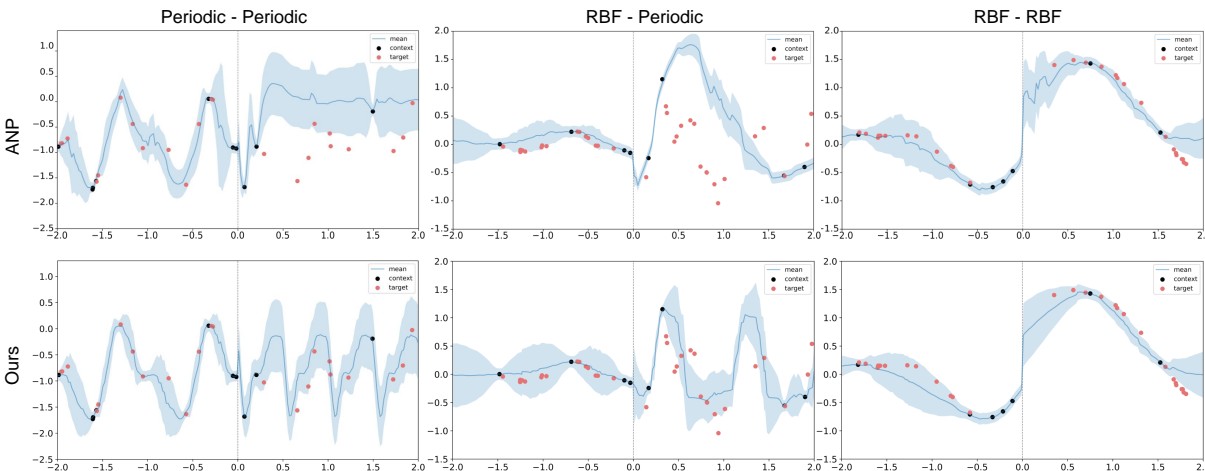

Figure 5: Comparison of predictions by ANP and DKNP. The dark blue line indicates the mean, and the light blue shade indicates one standard deviation range.

| Methods | Periodic - Periodic | RBF - Periodic | RBF - RBF |
|---------|---------------------|----------------|-----------|
| CNP     | -0.423 ($\pm$0.016) | 0.051 ($\pm$0.028)  | 0.966 ($\pm$0.006) |
| NP      | -0.531 ($\pm$0.022) | -0.007 ($\pm$0.011) | 0.872 ($\pm$0.017) |
| BNP     | -0.358 ($\pm$0.022) | 0.105 ($\pm$0.023)  | 0.984 ($\pm$0.011) |
| ANP     | 0.602 ($\pm$0.017)  | 0.883 ($\pm$0.005)  | 1.142 ($\pm$0.001) |
| BANP    | 0.605 ($\pm$0.026)  | 0.877 ($\pm$0.010)  | 1.145 ($\pm$0.001) |
| Ours    | **0.892** ($\pm$0.003) | **0.955** ($\pm$0.013) | **1.260** ($\pm$0.000) |

Table 2: Results (log-likelihood) of the NP variants and DKNP for three different change point configurations.

Figure 3, for example, clearly demonstrates the distinguishing feature of DKNP compared to previous NP variants, in this case ANP, which recorded the second highest likelihood in Table 1. In all kernel types, ANP fails to contain the target points within one standard deviation range while DKNP successfully captures the uncertainty in all unobserved points. Specifically, ANP is over-confident when modeling kernel types RBF and Matérn 5/2, where only a local structure exists. For periodic kernels, where both local and global structure exists, ANP fails to capture the underlying periodicity, due to its architecture that tries to estimate functional stochasticity with a fixed-size global latent variable. Unlike ANP, however, DKNP can correctly model the whole stochastic process thanks to its ability to explicitly model the correlations between all output variables.

Employing decoupled paths for modeling mean and covariance as in Figure 2, another advantage of DKNP is improved interpretability, as we can explicitly check the learned prior by visualizing the covariance matrix as described in Section 3. Such example is depicted in Figure 4, where we can readily compare the learned prior with the ground truth prior for all three kernel types. Note that DKNP not only allows us to visually check the learned prior, but also demonstrate its ability to accurately learn the ground truth kernels for all kernel types.

## 4.2   1D Regression with Change Point

Next we test all methods in a more challenging setup, where the underlying dynamics of the stochastic process changes at midpoint. Specifically, we employ three different GP configurations: Periodic-Periodic, RBF-Periodic, and RBF-RBF. Each configuration uses one kernel type (with a fixed hyperparamter setup) up to point 0, then another kernel type afterwards. As the correlations between output variables are more complex in this setup, we expect DKNP to demonstrate even more distinguishing performance than NP variants. The training and evaluation processes are the same as the previous experiments in Section 4.1. See Appendix C.1 for data generation details.

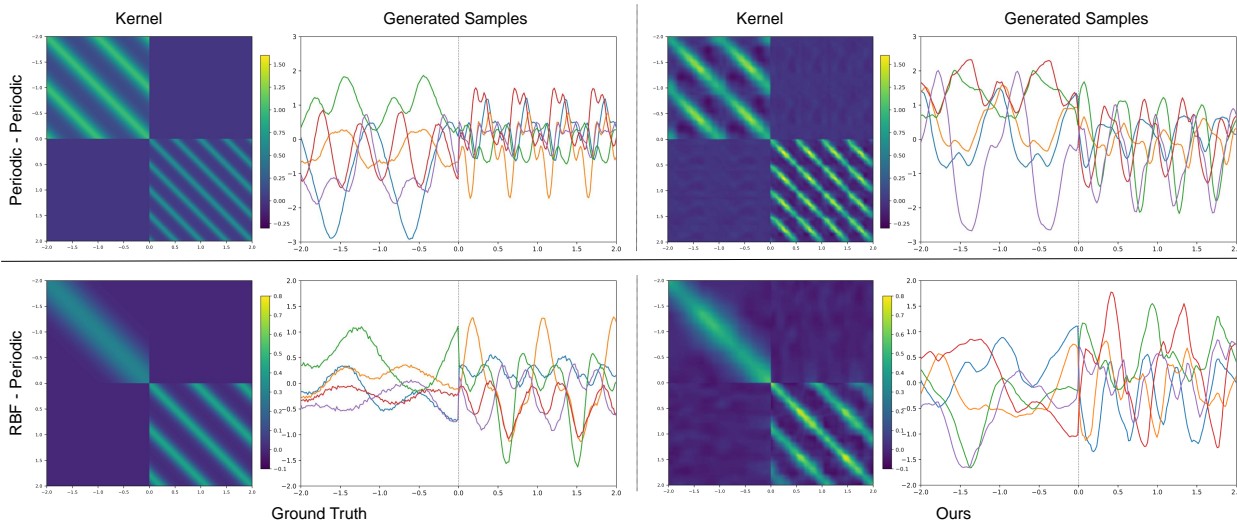

Figure 6: Visualization of the learned correlation matrices and the sampled data for Periodic-Periodic and RBF-Periodic compared to the ground truths.

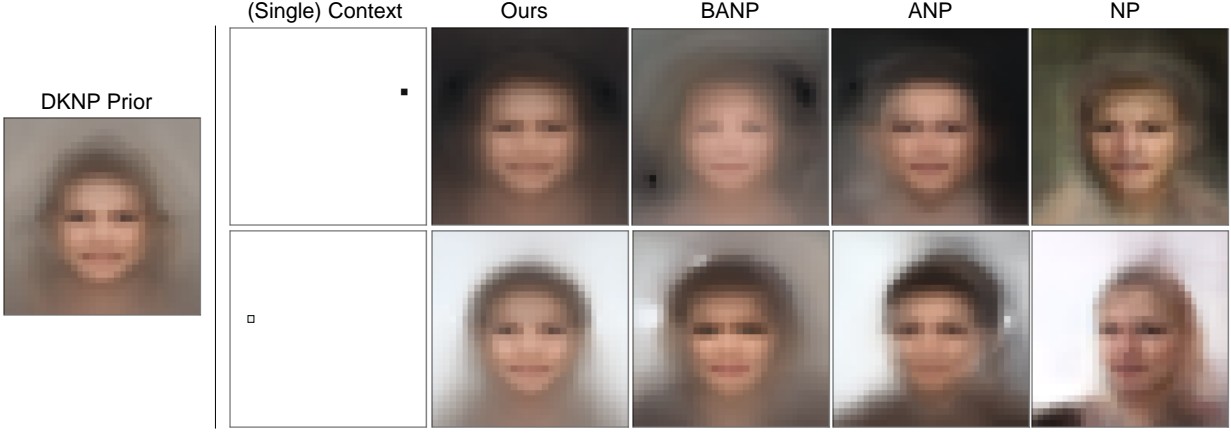

Figure 7: **(Left)** The prior learned by DKNP. **(Right)** Predicted means of various models given single context point shown in first column.

The results in Table 2, as expected, show wider gaps between DKNP and NP variants for all three configurations in terms of log-likelihood. Interestingly, unlike the the previous results (Table 1) where DKNP marginally outperformed NP variants for RBF kernels (but beyond one standard deviation margin), the performance gap is more distinguished even for RBF-RBF configuration in this setup. This observation indicates DKNP's ability to better capture complex correlations derived from a combination of rather simpler kernels than NP variants.

Figure 5 demonstrates predictions by ANP and DKNP for all three change point configurations. Compared to ANP which fails to correctly capture the changing dynamics of the given processes (especially for configurations including periodic kernels), DKNP shows its ability to correctly learn the underlying dynamics before and after the change point. In Figure 6, we compare the learned priors as well as the generated samples from them to the ground truths priors and samples for two change point configurations, Periodic-Periodic and RBF-Periodic. Note that DKNP can accurately learn two distinct kernels in both sides of the midpoint, (*i.e.* low and high frequency in the Periodic-Periodic case, smooth and periodic in the RBF-Periodic case) thus able to generate samples that are practically equivalent to the ground truth samples.

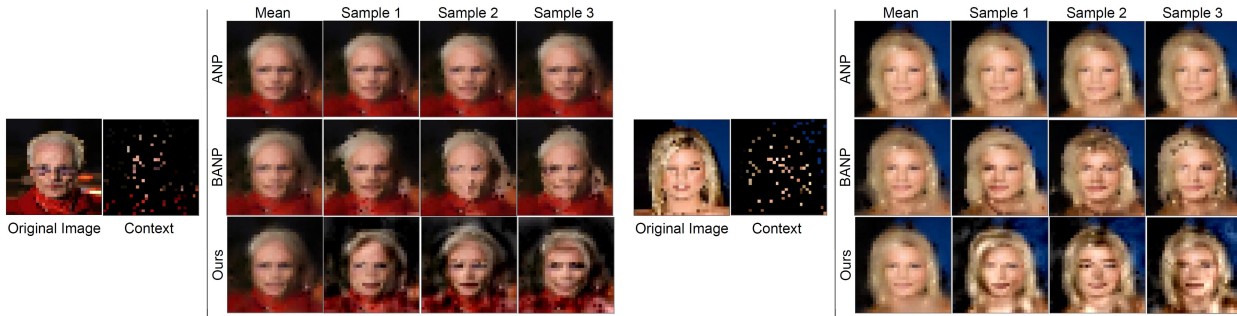

Figure 8: Examples of the image completion task. Based on the original image, roughly 10% of pixels (100 pixels) were given as the context. The filled images generated from DKNP (Ours) are much more diverse and clearer, compared to other baselines.

| Methods | Celeb A |
|---------|---------|
| CNP | 2.280 ($\pm$0.010) |
| NP | 2.374 ($\pm$0.024) |
| BNP | 2.876 ($\pm$0.014) |
| ANP | 3.471 ($\pm$0.005) |
| BANP | 3.627 ($\pm$0.001) |
| Ours | **3.951** ($\pm$0.021) |

Table 3: Results (log-likelihood) of NP variants and our model in the 2D image completion task.

### 4.3 2D Image Completion

The purpose of the 2D image completion task is to test how well each model shows its functional flexibility to learn non-trivial kernels. Assuming that image data is generated by a stochastic process, the task is to regress pixels that are missing based on the provided pixels (*i.e.* context pixels) in the image. Specifically, we use the CelebA dataset (Liu et al., 2015), which consists of 202,599 facial images of celebrities of diverse ethnicities, gender, and age.

For both training and evaluation, the number of context points $n_c$ is sampled from Unif(3, 200) and the target points are $n_c$ + Unif(3, 200 - $n_c$). Details about the data generation process, the training process, and the evaluation process are in Appendix C.2.

In Table 3, consistent with the previous results, DKNP shows superior ability in modeling even two-dimensional stochastic processes compared to all NP variants. This is somewhat predictable based on two previous results (Table 1 and Table 2) where the performance gap between DKNP and NP variants was more visible for stochastic processes with complex underlying dynamics (where both local and global structure exist), and the fact that the facial images are likely to follow non-trivial dynamics. For example, the dynamics to generate the hair or the background, which are typically low frequency signals, would be quite different from the dynamics to generate the details of the face where eyes, nose, and mouth altogether compose higher frequency signals. And unlike the previous change point data where the boundary between two different dynamics (*i.e.*, kernels) is a point in a 1D line, the boundary between different kernels in facial images is a (curved) line in a 2D plane (*e.g.* hairline). Therefore we can expect DKNP to significantly outperform all NP variants in estimating the correlations between output variables, as shown in Table 3.

On the left of Figure 7, we visualize the mean of the prior distribution, (*i.e.*, the output distribution that is predicted without any context point) learned by DKNP on the CelebA dataset. Note that DKNP's prior is purely data-driven based on the training data (facial images in this case) and can generate samples without encoding any context point, unlike the previous NPs that require the representation of context points to make

inference on target points. This figure demonstrates that DKNP learns the prior distribution appropriate for a given task and accurately captures the correlation between the output variables well.

The right part of Figure 7 illustrates the predictive mean of the NP variants and DKNP when a single context point is given in the background. The posterior mean of DKNP demonstrates distinct behavior for the face and background, such that the given context uniformly affects the background while keeping the face practically intact. This shows that DKNP accurately captures the correlations between output variables by successfully learning kernel functions that behave differently for face and background. BANP, ANP, and NP, on the other hand, demonstrate unpredictable behaviors where the context point in the background also affects the hair or the clothes, or the background is only partially affected by the given context point. This behavior reveals that NPs, which implicitly learn the kernel function, have a harder time modeling the accurate correlation between output variables.

We conclude this section with qualitative examples in Figure 8 that highlight the DKNP's capability to generate diverse samples thanks to its explicit modeling of the covariance. Given 10% of the data points as context, ANP samples are generated by sampling the latent variable $z$, BANP samples are generated via bootstrapping, and DKNP samples are generated with the predicted mean and covariance. While ANP, BANP and DKNP all produce reasonable predictive means, ANP's generated samples heavily resemble one another, whereas DKNP's generated samples demonstrate considerable diversity such as different genders, facial expressions and hair styles. BANP also demonstrates more varied samples than ANP thanks to its bootstrapping process, but they sometimes contain noisy pixels.

## 5  Related Work

Many advances in NPs are made with the advent of neural network-based stochastic processes such as NPs (Garnelo et al., 2018b) and CNPs (Garnelo et al., 2018a). ANP (Kim et al., 2019) show a dramatic performance gain by leveraging attention mechanisms in the aggregating operation. Also, Le et al. (2018) confirm through extensive empirical evaluation on the design of NPs such as architecture and objective functions that combining attention and NPs improve the predictive log-likelihood marginally. Lee et al. (2020) extends NPs using the bootstrap technique for estimating functional uncertainty without maintaining a latent variable in the NP architecture. This notion of removing the latent variable part is inline with our motivation.

For modeling *stationary stochastic processes*, Gordon et al. (2020) introduced Convolutional CNP (ConvCNP), where translation equivariance in the data is explicitly encoded in the model. Convolutional NPs (ConvNP) Foong et al. (2020) are a natural extension of ConvCNPs where a global latent variable was introduced to model dependencies of the predictive distribution. Gaussian Neural Processes (GNP) (Bruinsma et al., 2021) further generalize these classes of models with translation equivariance by leveraging convolutional neural networks to capture the predictive correlations. While GNPs share a similar motivation to our work, the way covariance matrix is parameterized is quite different (e.g., 1-D, 2-D convolution versus attention) from DKNPs, as their focus is mainly on translation equivariance. Specifically, the covariance matrix of GNPs depends on $Y_T$, while for DKNPs, motivated by posterior of GPs, the covariance only depends on the inputs $X_C$, $X_T$ (see equation 6). We report additional experimental results and discussion of these methods on 1-D regression in Appendix D. However, we note that the main goal of these works is modeling stochastic processes with stationary kernels by leveraging the translation equivariance property, which is orthogonal to our works.

DKNPs are also closely related to deep kernel learning (DKL) (Wilson et al., 2016b;a) in that DKNPs learn a kernel function through neural networks. However, DKNPs are different from DKL in the following aspects. First, as described in eqs. 5 and  6, DKNPs are motivated by GP posterior. In other words, DKNPs directly model the posterior distribution unlike DKL; i.e., with given context points, the outputs of DKNPs are the mean and covariance matrix of the posterior distribution. Such a characteristic allows DKNPs to have quadratic time complexity in the inference time since the posterior mean and covariance are obtained by a single forward propagation of the neural networks with attention modules. On the other hand, even though DKL uses neural networks to extract the representation vectors of the inputs similar to DKNPs, DKL still leverages a GP model. That is, without any approximation of a GP model, DKL has a cubic time complexity

for its inference since it requires the inverse of the covariance matrix to infer the GP posterior. Another difference is that DKNPs and DKL have different training regimes. Unlike GPs and DKL that are trained to learn a single function, NP variants including DKNPs are trained to learn multiple functions of stochastic processes.

## 6 Conclusion

We propose a new neural stochastic processes, Decoupled Kernel Neural Processes (DKNPs), that learn an explicit kernel function to better capture the correlation between output variables. By leveraging cross- and mixed attention mechanisms to model an explicit kernel function, DKNPs outperform the concurrent NP variants in terms of predictive likelihood and better global coherence of generated samples. By the novel model architecture of DKNPs, the learned prior can be accessible, which provides a deeper understanding of the underlying distributions of data. As future work, one could consider developing a method to manipulate a learned kernel or to impose a constraint on the kernel learning process of DKNPs with prior knowledge.

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

# A    Architectures Details

We mostly followed the same model architectures in the paper (Lee et al., 2020). *num_det_path* and *num_det_path* indicate the number of deterministic and stochastic paths.

## A.1    1D regression task

|  | CNP | NP | BNP | ANP | BANP | DKNP |
|---|---|---|---|---|---|---|
| dim_x | 1 | 1 | 1 | 1 | 1 | 1 |
| dim_y | 1 | 1 | 1 | 1 | 1 | 1 |
| dim_hid | 128 | 128 | 128 | 128 | 128 | 128 |
| dim_lat | - | 128 | - | 128 | - | - |
| num_det_path | 2 | 1 | 2 | 1 | 2 | 2 |
| num_stoch_path | 0 | 1 | 0 | 1 | 0 | 0 |
| enc_v_depth | - | - | - | 4 | 4 | 4 |
| enc_qk_depth | - | - | - | 2 | 2 | 2 |
| enc_pre_depth | 4 | 4 | 4 | 4 | 4 | 4 |
| enc_post_depth | 2 | 2 | 2 | 2 | 2 | 2 |
| dec_depth | 3 | 3 | 3 | 3 | 3 | 3 |

Table 4: Hyperparameter setting for the 1D regression task

## A.2    Image completion task

|  | CNP | NP | BNP | ANP | BANP | DKNP |
|---|---|---|---|---|---|---|
| dim_x | 2 | 2 | 2 | 2 | 2 | 2 |
| dim_y | 3 | 3 | 3 | 3 | 3 | 3 |
| dim_hid | 128 | 128 | 128 | 128 | 128 | 128 |
| dim_lat | - | 128 | - | 128 | - | - |
| num_det_path | 2 | 1 | 2 | 1 | 2 | 2 |
| num_stoch_path | 0 | 1 | 0 | 1 | 0 | 0 |
| enc_v_depth | - | - | - | 6 | 6 | 6 |
| enc_qk_depth | - | - | - | 3 | 3 | 3 |
| enc_pre_depth | 6 | 6 | 6 | 6 | 6 | 6 |
| enc_post_depth | 3 | 3 | 3 | 3 | 3 | 3 |
| dec_depth | 5 | 5 | 5 | 5 | 5 | 5 |

Table 5: Hyperparameter setting for the image completion task

# B    Computational Complexity

|  | CNP | NP | BNP | ANP | BANP | DKNP |
|---|---|---|---|---|---|---|
| 1D Regression | 2.60 | 3.54 | 5.74 | 5.89 | 9.18 | 5.84 |
| 2D Image Completion | 49.69 | 57.75 | 90.32 | 84.74 | 214.79 | 111.32 |

Table 6: Comparisons of the amount of training time for DKNPs and NP variants. We measure the time taken for 200 iterations for the 1D regression task and an epoch for the 2D image completion task.

According to the Eq. 7, DKNPs calculate the log-likelihood with full covariance matrix to optimize the objective function during the training. That is, the computational cost of DKNPs during the training time cubically increases along with the number of context and target points. However, NPs aim to address "few-shot function estimation" (Garnelo et al., 2018b;a) that predicts target points by adaptively estimating the function based on a few context points.

Since only a few context points are used, such cubical computational complexity brings marginal computational overhead compared to other baselines. Table 6 reports the amount of training time of DKNPs and NP variants on 1D regression and 2D image completion tasks. All the results are obtained from the dedicated server with the same hardware and software specifications. As shown, despite cubic computational complexity of DKNPs, the training time of DKNPs is comparable to other baselines and even more efficient in some cases. Yet, optimizing the training complexity of DKNPs is still a good direction of future work.

Additionally, DKNPs and GPs both have cubic computational complexity during the training time. However, GPs still have cubic complexity during the inference time (*i.e.,* predicting the mean and variance of target points) while DKNPs have quadratic complexity since DKNPs only require attention-based forward operations similar to ANPs.

## C   Experimental Details

### C.1   1D Regression

**Data generation from a single kernel**     We constructed training and test data generated from GPs with multiple kernels including RBF, periodic kernels, and Matérn 5/2. For each task, we generated $x \sim \text{Unif}(-2, 2)$ and $y$ from GP kernels where the GP hyperparameters are sampled from uniform distributions. For RBF kernel, $k(x, x') = s^2 \exp(-||x - x'||^2/2l^2)$, we sampled $s \sim \text{Unif}(0.1, 1.0)$, $l \sim \text{Unif}(0.1, 0.6)$, and output additive noise $\mathcal{N}(0, 10^{-2})$. Given $n$, the size of the context $C$ was drawn from $\text{Unif}(3, 47)$ and the size of the target is sampled from $\text{Unif}(3, 50 - n_c)$. For Matérn 5/2 kernel $k(x, x') = s^2(1 + \sqrt{5}d/l + 5d^2/(3l^2)) \cdot \exp(-\sqrt{5}d/l), (d = ||x - x'||)$, we sampled from $s \sim \text{Unif}(0.1, 1.0)$ and $l \sim \text{Unif}(0.1, 0.6)$. For periodic kernel, $k(x, x') = s^2 \exp(-2\sin^2(\pi||x - x'||^2/p)/l^2)$, we sampled from $s \sim \text{Unif}(0.3, 1.0)$, $l \sim \text{Unif}(0.6, 1.0)$, and $p \sim \text{Unif}(0.8, 1.0)$. We trained all models identically for 100,000 steps with training batch size of 100. We used Adam optimizer with initial learning rate of $5 \cdot 10^{-4}$ and decayed using cosine annealing scheme.

**Data generation from two kernels**     Similar to a single kernel case, data points are sampled from a GP but with two types of kernels, RBF and periodic. The half of the points ranging from $x = -2$ and $x = 0$ were sampled from one kernel and the rest from $x = 0$ to $x = 2$ were from another kernel. We generated samples from three scenarios: Periodic - Periodic, RBF - Periodic, and RBF - RBF. Note that we set the correlation between two kernels being zero. The training details are identical to the single kernel case.

### C.2   Image Completion

**CelebA32**     Similar to 1-D regression tasks, we randomly sampled pixels of a given image at training as targets, and treat a subset of the points as contexts. The size of the contexts and targets is sampled from $\text{Unif}(3, 200)$ and $n_c + \text{Unif}(0, 200 - n_c)$. For preprocessing, $x$ is rescaled to [-1, 1] and $y$ is rescaled to [-0.5, 0.5]. We trained all models identically for 200 steps with training batch size of 100. We used Adam optimizer with initial learning rate of $5 \cdot 10^{-4}$ with cosine annealing scheme.

## D   Additional Results on 1D Regression

In this section, we compare DKNPs to ConvCNPs and GNPs whose main focus is on modeling stationary stochastic processes by leveraging the translation equivariance property. We follow the same training protocols from Sec. 4.1 which trains three different types of stationary stochastic processes: GP with RBF, Periodic, and Matérn 5/2 kernel. We also conduct 1D regression experiments with change point configurations used in Sec. 4.2.

The results in Table 7.7 demonstrate DKNP outperforms ConvCNP-XL across all 1D regression tasks. GNP and our method show comparable results in all tasks. However, as also noted in the original GNP paper (Bruinsma et al., 2021), GNP has a severe scalability issue that limits the applicability of the model in high-dimensional scenarios, e.g., an image completion task that we conduct in Sec. 4.3.

| Methods | RBF | Periodic | Matérn 5/2 | Periodic - Periodic | RBF - Periodic | RBF - RBF |
|---|---|---|---|---|---|---|
| ConvCNP-XL | 0.970 (±0.013) | 0.670 (±0.023) | 0.887 (±0.017) | -0.240 (±0.055) | 0.609 (±0.609) | 0.962 (±0.017) |
| GNP | 1.187 (±0.000) | 1.045 (±0.003) | 1.117 (±0.001) | 0.879 (±0.005) | 1.002 (±0.006) | 1.228 (±0.002) |
| Ours | 1.109 (±0.001) | 0.941 (±0.006) | 1.039 (±0.004) | 0.892 (±0.003) | 0.955 (±0.013) | 1.260 (±0.000) |

Table 7: Results (log-likelihood) of the NP variants and DKNP in multiple 1D regression tasks

# E  Additional Results on 2D Image Completion

In Figure 9, we present additional results for the image completion task.

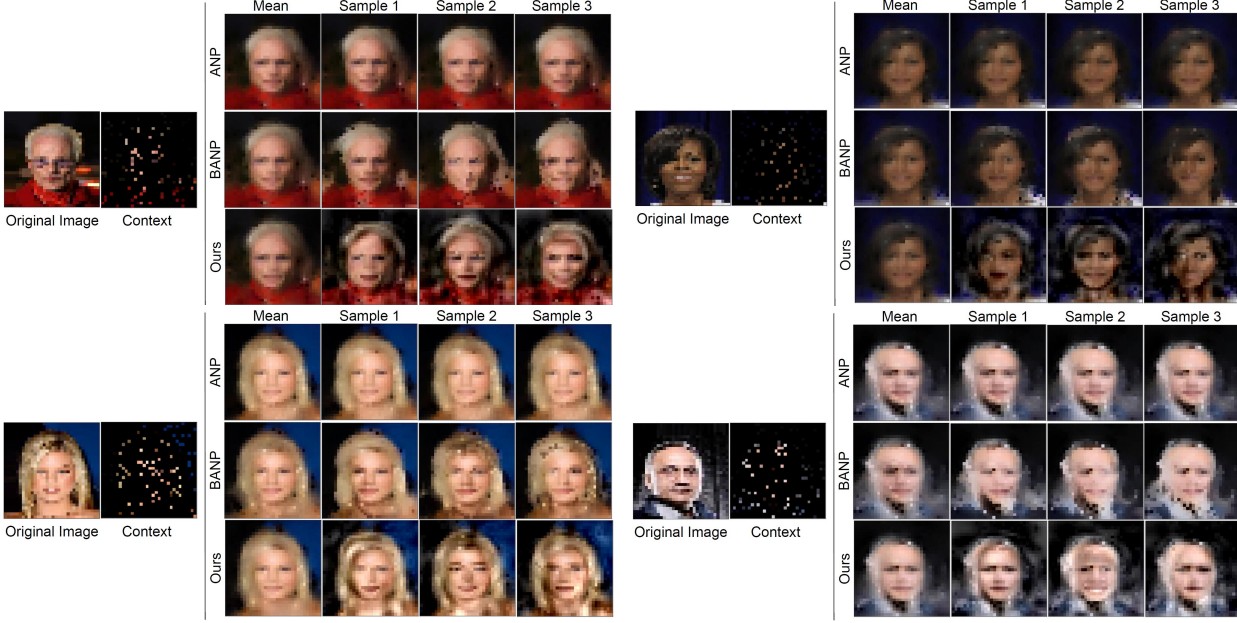

Figure 9: Examples of the image completion task. Based on the original image, roughly 10% of pixels (100 pixels) were given as the context. The filled images generated from DKNP (Ours) are much more diverse and clearer, compared to other baselines.

