# OpenReview forum: "Decoupled Kernel Neural Processes: Neural-Network-Parameterized Stochastic Processes Using an Explicit Data-driven Kernel"
_TMLR — Rejected by TMLR_

### Review · Reviewer_UKLw · 2022-06-14

**Summary Of Contributions:**

This paper proposes decoupled kernel neural processes (DKNP), a type of deep Gaussian process (GP) whose kernel is given by a deep neural network with an attention mechanism. The authors compare empirical performance against various types of neural processes (NPs) in two simulated-data tasks, and an image completion task, showing better performance than the considered baselines.

**Broader Impact Concerns:**

I have no broader impact concerns.

**Requested Changes:**

I consider the following changes completely necessary:

- Discuss DKNPs in the context of GPs with deep kernels, and either (1) convince me I am wrong and that DKNPs are not an instance of GPs with deep kernels, or (2) compare performance against GPs with a non-attention based kernel and show that DKNPs "blow them out of the water" across a variety of tasks and settings in terms of empirical performance. My rationale for asking this is as follows: if I am right and DKNPs are an instance of GPs with deep kernels, then the paper is not presenting anything insightful, and could really be summarized by "using attention is a good architecture for deep kernels", which would surprise no one and would be of little interest to the community, unless empirical results are exceptional.

- Empirically compare against GPs, both with and without deep kernels.

- Consider more tasks with non-toy settings, such as 1-d regression on a non-simulated dataset, or Bayesian optimization.

- Explain the complexity costs in more detail.

- Explain why DKNPs are consistent under marginalization.

- Improve the mathematical presentation.

**Strengths And Weaknesses:**

### Strengths:

- The problem considered in this paper, namely how to construct GP-inspired distributions over functions while taking advantage of advances in neural networks, is relevant.
- The empirical results in the paper seem to indicate the proposed method does outperform current neural processes.

### Weaknesses:
#### I believe this paper has several weaknesses, which I list below in an approximately decreasing order of importance from my point of view:

- The authors do not discuss deep kernels and their use in GPs, e.g. [1], and the many follow-ups. Not only is this line of work intimately related to DKNPs, but I think it actually admits DKNPs as a special case. DKNPs seem to me to be much more similar to GPs with deep kernels than they are to other NPs. In this view, DKNPs are just GPs with a deep kernel with a specific architecture leveraging attention, rather than a novel type of NP.

- No comparisons are carried out against GPs. NP papers often carry out these comparisons, and given the similarity of the proposed method to GPs, I found the lack of such comparisons to be a huge omission.

- The paper presents a single non-toy experiment on image completion.

- It is not immediately obvious that the proposed model (equation 6 and figure 2) defines a valid stochastic process. In order to be properly defined, a stochastic process over functions should obey permutation invariance, and be consistent under marginalization. The authors mention this in passing in page 5, implying that their model is consistent under marginalization, but I see this as a key point which should be fleshed out in much more detail. More precisely, let $T'$ be a set of indices obtained by removing a single index from $T$. Denote as $\Sigma(T)$ the $\Sigma$ that is defined on page 5, and let $\Sigma(T')$ be the analogous matrix corresponding to the covariance of $Y_{T'}|X_{T'}, X_C, Y_C$ (given by equation 6). In order for consistency under marginalization to hold, $\Sigma(T')$ has to be a submatrix of $\Sigma(T)$, obtained by removing the row and column corresponding to the index that was removed from $T$ to obtain $T'$. The authors should carefully explain, at the very least in the appendix, why $\Sigma(T')$ is indeed a submatrix of $\Sigma(T)$. To be clear, I believe this property (and permutation invariance) hold, but the authors should present this much more carefully.

- In the appendix, the authors claim their method has complexity $O(n(n+m))$ (which is sloppy notation, more on this in the next point). It is not clear to me why this is the case, and I would like to see a more detailed explanation of this. In particular, maximizing equation 6 should involve computing and backpropagating through the determinant of a $|T| \times |T|$ covariance matrix, which should have $O(|T|^3)$ complexity.

- The provided code has no README.

- The writing of the paper is rather sloppy, particularly the mathematical presentation. Below I write down examples of such sloppiness:

1. The title is missing a hyphen ("Neural-Network-Parameterized"), is inconsistent in its use of capital letters ("Data-Driven", and "Using"), and is grammatically incorrect ("Using an Explicit Data-Driven Kernel" or "Using Explicit-Data Driven Kernels").

2. The authors claim that the lack of kernels in NPs is limiting (both in the abstract and in the main manuscript). This is not immediately obvious, and seems like an intuition rather than a substantiated claim. An example given by the authors is periodic behaviour, but there is no conceptual limitation as to why periodic information could not be captured by a finite-length latent variable. This also seems contradictory to the claim that "NPs implicitly learn [...] the covariance of the output variables".

3. "maintaining explicit kernel functions, a key component of stochastic processes" is not technically correct, many stochastic processes do not rely on kernels (e.g. Poisson processes).

4. The presentation of NPs in section 2.1 feels backwards: the authors first explain the encoder, and then the decoder. This amounts to explaining how to do variational inference in a model before defining what the model is. Also, $T$ and $C$ are not defined, and the reader has to either understand them from context, or be familiar with common NP notation. This is particularly confusing, as equation 2 only conditions $Y_T$ on $X_T$ and not $X_C$, and it is only later explained that $X_C \subset X_T$. Similarly, $f_\theta$ is not properly defined (from where to where it maps should be explicitly written), and the same goes for $g_\phi$. The notation $g_\phi([x_i;y_i])$ is also inconsistent with its use of brackets with the rest of the paper. Also, $\sigma_z$ is defined as the output of a neural network, and I suspect that the output is actually $\log \sigma_z$.

5. Similarly, the mathematical notation in section 2.2 is sloppy. Attention is defined in equation 3, but the sizes of the matrices now use $n$, $d_{model}$, and $m$, which have not been defined (if I understood correctly, $n=|C|$ and $m=|T \setminus C|$, but notation should be made consistent between sections). Equation 3 also uses a softmax over matrices but does not specify the dimension along which the softmax is applied (I am aware this is common of papers using attention, but it does not excuse it, particularly for a journal paper). The matrices $W_s^K, W_s^V$, and $W_t^Q$ are similarly not defined, and neither is the source $S$. The subindex $i$ is used to index attention heads and datapoints, which again, is confusing notation. Equation 4 uses a concatenation operation but does not define the dimension along which the concatenation occurs, and it also does not define $W^O$. The attention score $a_i$ is also mentioned on page 4 but not defined.

6. Equations 5 and 6 are hard to read, I would recommend using colours (or more spacing) to make it easier to identify the mean and the covariance in each model.

7. The authors claim "This decoupled process allows DKNPs to explicitly learn the prior of the given dataset", which is technically incorrect, as this "learned prior" depends on $X_C$. One could do the same with a GP and obtain its conditional distribution given $X_C$.

8. Additional question: In figure 7, why does the leftmost image (from the DKNP prior) not match the second (from the left) bottom image on the right panel (i.e. the "Ours" with no context given)? Is the left image a sample, and the other just the mean?

9. Awkward/incorrect phrasings throughout the paper:

"capturing the underlying distributions can be restricted in complex scenarios"

"after learning samples generated from..."

"DKNP is beneficial for modeling the distribution having complex kernel functions"

"Given a function from stochastic process observed in $n$ points..."

"For a set of target input $X_T$..."

"For obtaining the predictive distribution $\log p(y_i|x_i, X_C, Y_C)$" (the predictive distribution has no log)

$(X_C, Y_C)$ is a tuple of sets, not a properly defined set, so saying "$(x_i, y_i) \in (X_C, Y_C)$", while understandable, is incorrect

"represents each context points in"

in "$p(y_t|x_t, r_C, z) = \mathcal{N}(y; \mu_y, \sigma^2_y)$", $y$ should be $y_t$

the notations "$q(z|e_T)$" and "$q(z|X_T, Y_T)$" are used interchangeably, which while correct is confusing

"the distribution of $q(z|X_C, X_T)$ captures the..." is incorrect, $q$ is itself the distribution, it has no distribution

"which is demonstrated with the coherent sample generation" is not clear

"Specifically, the DKNPs pass each context point {$(x_i, y_i)$}, the concatenation of $x_i$ and $y_i$ to MLP_\theta and represent it as $e_i$"

"for all inputs $(x_i)_{i \in C \cup T}$ ..." is also weird notation if assuming that $C \subset T$

"This allows to do inference the output distributions ..."

"prior distributions of DKNPs|the learned..."

"Different from the NPs..."

"the lower bound of conditional log-likelihood ... or its slight modification"

"that DKNP not only model allows us"

"202,599 number of facial images..."

"target points is $n_c$ + ..."



[1] Deep Kernel Learning, Wilson et al., AISTATS, 2016

---

> ### Author Response · Authors · 2022-07-06
> **Response to Reviewer UKLw**
>
> ### 1. Relation with Deep kernel learning
> We agree that DKNPs are related to deep kernel learning (DKL) in that DKNPs learn a kernel function through neural networks. However, DKNPs are different from DKL in the following aspects. First, DKL leverages both neural networks and GPs, obtaining the representation vectors of the data using the neural networks and applying GPs to those vectors. Thus, DKL still necessitates GPs, requiring user-specified and explicit kernel functions. Additionally, DKNPs and DKL have differences in training regimes. Unlike GPs and DKL which are trained to learn a single function, NP variants including DKNPs are trained to learn multiple functions of stochastic processes. Such differences are also described in existing studies [1, 2], and we added this to the related work of the manuscript.
> In this regard, directly comparing GPs and NP variants is not plausible. Instead, we added Oracle GP to Table 1 to provide the upper bound of ours and baselines.
>
> ### 2. Computational complexity
> As the reviewer pointed out, during the training time, DKNPs have computational complexity of $\mathcal{O}(T^3)$ where $T$ indicates the number of target samples. However, considering that NPs are proposed to address the problem of few-shot function estimation, such cubic computational complexity only requires a trivial amount of computational overhead compared to the existing baselines. As shown in the table below, the training times of DKNPs are comparable to or even lower than other baselines.
>
>
> We added this discussion to Appendix B in the manuscript.
> |                     | CNP   | NP    | BNP   | ANP   | BANP   | DKNP   |
> |---------------------|-------|-------|-------|-------|--------|--------|
> | 1D Regression       | 2.60  | 3.54  | 5.74  | 5.89  | 9.18   | 5.84   |
> | 2D Image Completion | 49.69 | 57.75 | 90.32 | 84.74 | 214.79 | 111.32 |
>
>
> ### 3. Consistency under marginalization
> Thank you for pointing this out. We agree that we need more explanations for consistency under marginalization. We added more explanations and details in Section 3.
>
> ### 4. README
> We newly added the README file and explained the detailed instructions in the file.
>
> ### 5. Figure 7
> We apologize for the confusion. The second (from the left) bottom image on the right panel is not a “no context” image. It contains one light-colored pixel on the left side of the image. For a better visibility, we added borderlines to the pixel and updated the figure.
>
> ### 6. Writing of the paper
> We deeply appreciate your detailed feedback. We faithfully reflected your comments in our manuscript.

---

> > ### Author Response · Authors · 2022-07-06
> > **Response to Reviewer UKLw**
> >
> > [1] Garnelo et al., Neural Processes, ICMLW, 2018
> >
> >
> > [2] Kim et al., Attentive Neural Processes, ICLR, 2019

---

> > ### Comment · Reviewer_UKLw · 2022-07-10
> > **Reply**
> >
> > I thank the authors for their reply. I have gone over the modifications to the manuscript, which address several of the clarity points I raised in my review. I am still not convinced about the difference from deep kernels however: looking  at equation 7 and defining the kernel $k(x,x') = x^\top x'$ and the neural network $w(x; X_C)$ given by applying the covariance function on equation 6 to the input $x$, the resulting covariance matrix can be written as $\Sigma_{ij} = k(w(x_i; X_C), w(x_j; X_C))$, as noted in the discussion above equation 7. How is this not directly specifying the kernel $k$ manually, and applying it to the outputs of a neural network?

---

> > > ### Comment · Reviewer_UKLw · 2022-07-18
> > > **Reminder**
> > >
> > > I kindly remind the authors that the discussion phase will end soon and my question above remains unanswered. Could you please elaborate on why you believe the proposed method cannot be interpreted as GPs with deep kernels given by the kernel described in my previous comment? Could you also further address the point raised by reviewer mBLp about what the main takeaway message of the paper is, particularly in the context of GPs and deep kernel learning?

---

> > > ### Author Response · Authors · 2022-07-20
> > > **Response to Reviewer UKLw**
> > >
> > > We deeply appreciate your insightful comment. We agree with your point that DKNPs could be interpreted as utilizing the inner product kernel. Accordingly, we removed the claim related to your point from the manuscript in Section 5. Specifically, there were two differences between DKNPs and DKL claimed in the manuscript; 1) a user-specified kernel is not required, and 2) the training regime is different, and we removed the former one. Instead, we additionally elaborated on the difference between them from another perspective.
> > >
> > > As described in eqs. (5) and (6) in the manuscript, DKNPs are motivated by GP posterior. In other words, DKNPs directly model the posterior distribution unlike DKL; i.e., with given context points, the outputs of DKNPs are the mean and covariance matrix of the posterior distribution. Such a characteristic allows DKNPs to have quadratic time complexity in the inference time since the posterior mean and covariance are obtained by a single forward propagation of the neural networks with attention modules.
> > >
> > > On the other hand, even though DKL uses neural networks to extract the representation vectors of the inputs similar to DKNPs, DKL still leverages a GP model. That is, DKL has a cubic time complexity for its inference since it requires the inverse of the covariance matrix to infer the GP posterior.
> > >
> > > Again, we deeply appreciate your precise reviews, and we faithfully reflected your points in the updated manuscript.

---

> > > > ### Comment · Reviewer_UKLw · 2022-07-21
> > > > **Follow-up question**
> > > >
> > > > I thank the authors for their reply. I have read the reply to my question, as well as the one responding to reviewer mBLp, and I am still not convinced the proposed method is that different from deep kernel learning: While I agree with the authors that the training regime is indeed different due to the use of context points, given the point above about the proposed method allowing the interpretation of applying a kernel to the output of a neural network, I believe one can also just interpret the model as a GP whose deep kernel is learned from $X_C$. Could you please further comment on this?

---

> > > > > ### Author Response · Authors · 2022-07-22
> > > > > **Response to Reviewer UKLw**
> > > > >
> > > > > We appreciate your comments, and we are happy with your agreement on the point that there is a difference between DKNPs and GPs in the training regime.
> > > > >
> > > > > Even though DKNPs could be interpreted as applying an inner product kernel to the output of neural networks, we want to emphasize that the output of the inner product kernel in DKNPs is ***an element in the covariance matrix of the posterior distribution*** with given context points.
> > > > >
> > > > > However, in the case of DKL, the output of a kernel function is ***an element in the covariance matrix of the prior distribution***. That is, to calculate the covariance matrix of the posterior distribution, DKL necessitates the inversion of matrices, which requires heavy computational costs.
> > > > >
> > > > > Furthermore, since DKNPs are composed of two separated paths in the architecture, the obtained covariance matrix (i.e., lower path) is not directly used to calculate the mean vector (i.e., upper path), which is different from DKL or GPs.
> > > > >
> > > > > We agree that DKNPs (including the existing NP variants) and DKL are closely related to each other. However, we still believe that there are differences in their mathematical formulations, architectures, and applicability between DKNPs and GPs with deep kernels.

---

### Review · Reviewer_Y5gj · 2022-06-20

**Summary Of Contributions:**

The paper proposes an alternative parameterization of a neural process that can explicitly model a separate mean and kernel function using attention mechanisms. The new parameterization allows to inspect learned kernels, which might help interpretability, and performs better than prior methods on several benchmarks.

**Broader Impact Concerns:**

Technical contributions, I have no concerns.

**Requested Changes:**

- Figure 1: very hard to read legend, axis ticks, and compare the functions. Font sizes should match the template better.
- Figure 2: make clear what part of the attention model is new compared to ANPs either in the caption or in the figure itself.
- The overall training process is not 100% clear to me. It would be great if it could be made more precise and discuss the complexity incurred. From the objective in Eq. (7) it seems like you would need to invert a covariance matrix of $T \times T$ dimensions with $T$ training data points? Is that the case? Further, the formulation in Eq. (7) reminds me of inducing point Gaussian process posteriors. So maybe this is in fact similar to a lower-bound when treating conditioning points as latent variables. Just an idea/analogy to consider..
- Typos:
  - "function from [a] stochastic process"
  - "represents each context point[s]"
  - "This allows to do inference the output prior of DKNP"
  - "distributions of DKNP|the learned prior"?
  - "between all output variableS"

**Strengths And Weaknesses:**

### Strengths
- improved performance in complicated settings like image generation and stochastic processes with change points
- overall simple and apparently effective algorithm, useful amendments to prior approaches
- derivation and motivation are clear, the form of the predictive is inspired by GP inference
- possibility to inspect the learned covariance matrix
- DKNPs prior is data driven and can generate samples without context, very impressive in the CelebA experiment
- sampling diversity, as claimed, seems superior to BANP and ANP models

### Weaknesses
- No discussion of computational complexity. Modelling a full covariance should be significantly more complex than modelling a latent variable $\mathbf{z}$ only
- Table 1 should also contain the attainable ground truth log likelihood for the corresponding true model the data is generated from. This allows to quantify the gap introduced by NP methods and show relative improvement towards the ground truth
- It is not clear to me how Eq. (7) is equivalent to CNP's objective. From the background, I understand it is Eq. (2) that is commonly used. Eq. (7) resembles the maximum log predictive likelihood. In comparison to CNPs, is the idea to model that directly as opposed to via $\mathbf{z}$?
- Is it possible to obtain a representation of the kernel for other NPs by sampling? In the end it's just a covariance estimation problem and I wonder if this wouldn't be an alternative if one wanted to obtain it.

---

> ### Author Response · Authors · 2022-07-06
> **Response to Reviewer Y5gj**
>
> ### 1. Discussion of computational complexity
> As reviewer 2 pointed out, DKNPs calculate the full covariance matrix to obtain likelihood. Thus, unlike existing NP variants that utilize latent variable $z$, DKNPs have a computational complexity of $\mathcal{O}(T^3)$ during the training time where T denotes the number of target points.
>
> However, since NP variants including DKNPs aim to address the problem of few-shot function estimation, the actual computational overhead of DKNPs during the training time is marginal as shown in the table below. According to the results in the table, DKNPs show comparable or more efficient training time compared to other baselines.
>
> We added the results and discussion to Appendix B in the manuscript.
> |                     | CNP   | NP    | BNP   | ANP   | BANP   | DKNP   |
> |---------------------|-------|-------|-------|-------|--------|--------|
> | 1D Regression       | 2.60  | 3.54  | 5.74  | 5.89  | 9.18   | 5.84   |
> | 2D Image Completion | 49.69 | 57.75 | 90.32 | 84.74 | 214.79 | 111.32 |
>
>
> ### 2. ground truth log-likelihood in Table 1
> As the reviewer proposed, we added the results of oracle GP to Table 1 in our manuscript.
>
>
> ### 3. Eq. (7) is equivalent to CNP's objective
> Eq. (2) describes the objective function of NP that utilizes the latent variable $z$ to encode context points. CNPs also encode context points into a single vector, but they do not leverage a latent variable. Thus, the objective function of CNPs is the log-likelihood of the normal distribution, which is the same as DKNPs.
>
>
> ### 4. Is it possible to obtain a representation of the kernel for other NPs by sampling?
> Yes, we can estimate the covariance matrix for other NPs by sampling. However, we conjecture that the covariance matrix obtained from sampling may improve the interpretability of the model but is difficult to be used for the inference. Thus, utilizing the covariance matrix obtained from sampling has limited applicability compared to our DKNPs.
>
>
> ### 5. Requested changes (figure 1, caption of figure 2, computational complexity, typos)
> Thank you for your suggestions. We reflected your points in our manuscript.

---

### Review · Reviewer_mBLp · 2022-06-23

**Summary Of Contributions:**

The authors propose a multihead attention architecture for learning the mean and covariance functions in a Gaussian process regression model. This allows for a Gaussian process regression model with data-driven kernel and mean. The authors conclude with some toy experiments.

**Broader Impact Concerns:**

No significant broader impact concerns.

**Requested Changes:**

There are a few unsupported claims and issues in presentation. Each is minor, but their total sum is not insignificant.
- Figure 1 caption. "As a result of kernel learning in the DKNP, the underlying prior distribution of data can be inferred (periodic
kernel in this case)." - Is this true in general? I can see how this can be inferred in this particular example, but not in general. Perhaps this statement should be switched to "as a result of kernel learning in the DKNP, in this case the periodic prior distribution of data can be inferred." Similar the following statement is not supported by evidence "Also, as visualized in Figure. 1(b), DKNP achieves improved interpretability via explicitly learned kernels and better global coherence of generated samples."

- Abstract. "While this data-driven learning of stochastic processes has been shown to model various types of data...". What does this mean? Using mode as a verb usually refers to models in a traditional scientific sense, with associated interpretability etc. I don't think this is what is meant by the authors.

- Equation (5) looks like the posterior predictive distribution for "vanilla" Gaussian process regression i.e. y_i = f(x_i) + \epsilon_i, with the various assumptions on \epsilon (It looks like there is no noise here). However the paper just states that this is the posterior predictive distribution of GPs, without reference to likelihood or model. The words could also be used to refer to the posterior predictive distribution of a GP for classification (which does not have a closed form). Please be more precise. Is there a reason why we do not model noise \epsilon in (5)?


Questions:
- What is the relationship between the text in 2.1 with the symbols in figure 2? Maybe none, since figure 2 relates to section 3? But some of the symbols are overloaded, being used in 2.1 and 3 to represent different things. Is there any way to not overload these symbols?
- Footnote 2. Equation 5 and 6 are obviously different! I guess you are trying to say something a bit more subtle here, maybe something to do with the way they depend on C and T? Can you try and rephrase?
-  Can you clarify what this sentence means in the conclusion? "By the novel model architecture of DKNPs, the learned prior can be accessible, which provides a deeper understanding of the underlying distributions of data." By having a learned prior be accessible, are you referring to plots like Figure 4? How does this provide a deeper understanding?


Minor:
- Section 2.2. I'm having trouble understanding "The attention mechanism can be calculated from multiple subspaces, namely, multihead attention (MHA)". Are you saying that multihead attention are the multiple subspaces? Or multihead attention allows calculation from multiple subspaces? It reads as the former at the moment.
- Footnote 1. It is not "the" typical setting for GPs, but "a" typical setting.


**Strengths And Weaknesses:**

Strengths
- The authors investigate the use of an attention based neural network architecture for the mean and covariance function of a Gaussian process regression model. The story is relatively well-sold and easy to follow. The idea is solid and worth investigating if it can generate insights into data-driven kernels for Gaussian process models.

Weaknesses
- I am having some trouble determining what generalisable insights are surfaced by this paper. Others have considered learned kernels and mean functions in the context of Gaussian processes or more generally kernel methods. Probably not much work exists on using attention mechanisms to model kernels and means, but apart from introducing this very specific neural network configuration, I'm not sure what the reader is supposed to take from this paper.
- I don't understand what the reader should glean from figures 7 and 8. Can you elaborate on this? I cannot agree with the statement "The filled images generated from DKNP (Ours) are much more diverse and clearer, compared to other baselines." based on the images provided.
- There are some unsupported claims (detailed in "requested changes").

---

> ### Author Response · Authors · 2022-07-06
> **Response to Reviewer mBLp**
>
> ### 1.  Generalizable insight
> There exist major differences between GP models and NP variants including DKNPs. First, GPs are usually trained to fit a single function, while NP variants are trained with samples generated from multiple functions. Furthermore, unlike GPs, NP variants including DKNPs do not require user-specified kernel functions. In this regard, our proposed method, DKNPs, has to be considered as a variant of NPs rather than one of GP regression models or kernel methods.
>
> The main insight we provided was the importance of the full covariance matrix to capture the correlations between the output variables. By leveraging it, we achieved better quantitative results and interpretability compared to the existing baselines that utilized a fixed-length latent variable $z$ to capture the dependencies of output variables.
>
> ### 2. Figures 7 and 8
> As elaborated in Section 4.3 and Fig. 7, DKNP is the only model that shows distinct behavior on the background and face when a single context point is given. That is, DKNPs more effectively capture the correlations between the output variables compared to other NP variants by leveraging the full covariance matrix.
> In Fig. 8, we intended to show that DKNP generates more diverse images than ANP and clearer images compared to BANP. We will revise this in our final version. Additionally, as shown in Table 3, we want to emphasize that DKNPs outperform the baselines in terms of the log-likelihood metric.
>
> ### 3. Figure 1 caption
> Based on the results of Figs. 4, 6, and 7, we believe that DKNPs reasonably infer the underlying prior distribution of data in general.
>
> ### 4. learned prior and “deeper understanding”
> For the existing NP variants, it is not feasible to check which stochastic processes are learned after the training. For example, even after training the existing NPs with data sampled from the RBF kernel, there are no means to check the model indeed learned the RBF kernel. On the other hand, it is possible in the case of DKNPs by visualizing the learned kernel as shown in Fig. 4. Therefore, it provides a deeper understanding of the model, which was not feasible in the existing NP variants.
> ### 5. abstract
> We used the term ‘model’ as a meaning of “capturing the relationships between the input and output variables of the dataset by training”.
>
> ### 6. "vanilla" Gaussian process regression
> We specified the GP regression to Eq. (5) in the manuscript. We simplified Eq. (5) not to have \epsilon since we introduced Eq. (5) to explain the motivation of our method, DKNPs.

---

> > ### Comment · Reviewer_mBLp · 2022-07-18
> > **Discussion**
> >
> > Thanks for answering my queries and for your updates. Overall, I found the paper disappointing in two respects:
> >
> > 1. Unsupported and/or imprecise/sloppy claims. These are scattered throughout the paper. I picked out a few of these in my review, but it seems like Reviewer UKLw did a much more comprehensive job of this. Some of these have already been fixed. These claims can almost certainly be fixed without seriously affecting the main story or message of the paper. I am not satisfied with the fixes that have been done so far, e.g. on point 2 above, I don't feel there is sufficient evidence to claim that `` DKNP generates more diverse images than ANP and clearer images compared to BANP".
> >
> > 2. What is the story of the paper? If I were to summarise, I would say that the authors look at the idea of a very specific type of learned prior mean and covariance function for a Gaussian process. They authors mention the generalisable insight in their rebuttal above, " the importance of the full covariance matrix to capture the correlations between the output variables". But I am not able to find this insight easily in the paper, and I was not aware that this was the main message of the paper after reading. Unless new mathematical analysis is introduced and/or extensive, rigorous experiments are done, I don't find the story compelling. To be clear, I do not expect a new mathematical analysis or experiment to necessarily uncover a model that achieves amazing predictive performance.

---

> > > ### Author Response · Authors · 2022-07-21
> > > **Response to Reviewer mBLp**
> > >
> > > ### 1. Main story of the paper
> > > Our main story is as follows.
> > >
> > > As written in the second paragraph of Sec. 1, Neural Processes (NPs) have evolved to better capture the correlations between the output variables using a vector representation. For example, [1] utilized a latent variable for capturing the correlations, [2] improved the latent variable by using an attention mechanism, and [3] leveraged bootstrapping technique to achieve the same purpose.
> > >
> > > Unlike those approaches, DKNPs explicitly utilize the covariance matrix rather than using a form of vector representation. Although the theoretical analysis is not provided in our paper, we validate our intuition and the effectiveness of DKNPs by conducting experiments on three different tasks as shown in Tables 1, 2, and 3. As shown, DKNPs have significantly improved the log-likelihood compared to the existing NP baselines. Furthermore, as illustrated in Figs. 4, 6, and 7, DKNPs enable the visualization of the learned kernel functions, which was not feasible for the existing NP variants.
> > >
> > > Finally, we believe it is difficult to regard DKNPs as a specific type of GPs or DKL rather than one of the NP variants for two reasons. First, as described in eqs. (5) and (6) in the manuscript, DKNPs are motivated by GP posterior. In other words, DKNPs directly model the posterior distribution unlike GPs and DKL; i.e., with given context points, the outputs of DKNPs are the mean and covariance matrix of the posterior distribution. Additionally, DKNPs have a different training regime compared to GPs and DKL. Unlike GPs and DKL which are trained to learn a single function, NP variants including DKNPs are trained to learn multiple functions of stochastic processes. Due to such differences, we believe DKNPs have to be regarded as a particular type of NP variants.
> > >
> > > [1] Garnelo et al., Neural Processes, ICMLW, 2018
> > >
> > > [2] Kim et al., Attentive Neural Processes, ICLR, 2019
> > >
> > > [3] Lee et al., Bootstrapping Neural Processes, NeurIPS 2020
> > >
> > > ### 2. Figure 1
> > > As the reviewer pointed out, we edited the caption of Fig. 1 to be more specific.
> > >
> > > ### 3. Equations 5 and 6
> > > As the reviewer pointed out, eqs. (5) and (6) are clearly different. From the eqs. (5) and (6), we wanted to emphasize that DKNPs are motivated by the GP posterior distribution. Specifically, we intentionally arranged the two equations sequentially to show that the mean is obtained using X_T, X_C, and Y_C, and the covariance is obtained using X_T and X_C, in both GP posterior and DKNPs.

---

### Decision · Action_Editors · 2022-08-10

**Recommendation:** Reject

**Comment:**

The paper introduces a new deep learning approach that looks to model the posterior distribution of a Gaussian process. In particular, the authors use neural networks with an attention mechanism to directly model the mean and kernel function showing some empirical evidence that the approach outperforms previous neural processes variants based on encoding correlations in the output variables that use latent vectors alone. The intuition behind this paper is insightful and worth pursuing in the context of some of the relevant literature. TMLR editorial decisions (see https://www.jmlr.org/tmlr/ae-guide.html) are based on answers to the following questions: 1. "Are the claims made in the submission supported by accurate, convincing and clear evidence?" and "2. Would at least some individuals in TMLR's audience be interested in the findings of this paper?". Unfortunately, the current state of this paper is lacking in the answers to both questions. Reviewers mBLp and UKLw have pointed out how the paper lacks in both aspects, and I want to bring up some of their points here. Regarding "claims made", I agree with reviewer mBLP when they dispute the claim: "DKNP generates more diverse images than ANP
and clearer images compared to BANP". This seems to be the case in the context of a very specific example and would require a further, more thorough empirical validation; otherwise, it amounts to cherry-picking. I would suggest the authors check the literature on deep generative models where there is a large amount of research on empirically validating subjective results like the ones the authors are trying to analyse.  Incidentally, some claims in the OpenReview discussion need careful consideration: "Unlike GPs (...) which are trained to learn a single function". This is technically incorrect, GPs learn a distribution over functions, so they learn many functions, not one. The fact that a large portion of the literature focuses on learning a single function does not invalidate that one can use several functions (assuming they come from the same distribution) to learn the GP. Now, on the second point on "TMLR's audience", I agree with reviewer UKLw that the connections with DKL are still unclear or require further discussion.  Adding to this, the lack of a more thorough analysis of the computational complexity, the potential audience for this paper is still missing important messages.